# FootNet v1.0: Development of a machine learning emulator of atmospheric transport

Tai-Long He[1,a,*], Nikhil Dadheech[1,*], Tammy M. Thompson[2], and Alexander J. Turner[1]

[1]Department of Atmospheric and Climate Science, University of Washington, Seattle, WA, USA
[2]Environmental Defense Fund, Boulder, CO, USA
[a]now at: School of Engineering and Applied Sciences, Harvard University, Cambridge, 02138, USA
[*]These authors contributed equally to this work.

**Correspondence:** Alexander J. Turner (turneraj@uw.edu)

**Abstract.** There has been a proliferation of dense observing systems to monitor greenhouse gas (GHG) concentrations over the past decade. Estimating emissions with these observations is often done using an atmospheric transport model to characterize the source-receptor relationship, which is commonly termed measurement "footprint". Computing and storing footprints using full-physics models is becoming expensive due to the requirement of simulating atmospheric transport at high resolution.

We present the development of FootNet, a deep learning emulator of footprints at kilometer scale. We train and evaluate the emulator using footprints simulated using a Lagrangian particle dispersion model (LPDM). FootNet predicts the magnitudes and extents of footprints in near real-time with high fidelity. We identify the relative importance of input variables of FootNet to improve the interpretability of the model. Surface winds and a precomputed Gaussian plume from the receptor are identified to be the most important variables for footprint emulation. The FootNet emulator developed here may help address the

computational bottleneck of flux inversions using dense observations.

## 1 Introduction

Monitoring anthropogenic greenhouse gas (GHG) emissions is important for ensuring the success of the Paris Agreement's long-term goal on mitigating climate change (IPCC, 2022). To that end, there has been a proliferation of dense observing systems over the past decade to better track GHG emissions. Substantial efforts have been made to expand observation networks

to better quantify urban GHG emissions, as the majority of the world population lives in urban areas and the degree of urbanization is projected to increase in the future (United Nations Publications, 2019). For example, the Northeast Corridor GHG observation network was established to quantify emissions of carbon dioxide and methane using tower-based *in situ* measurements in urban regions in the northeastern United States (Karion et al., 2020). The BErkeley Atmospheric $CO_2$ Observation Network (BEACO$_2$N; Shusterman et al., 2016) utilizes low-cost sensors to increase the spatial density of measurements, which

could be used to estimate urban emissions on intra-city scales in the San Francisco (SF) Bay Area. The proliferation of urban GHG observation networks allows for decadal analyses of GHG emissions and provides information to improve the efficiency of GHG reduction policies (Mitchell et al., 2018; Lauvaux et al., 2020). There has been a coincident expansion in space-borne GHG monitoring instruments, which provide similarly dense observations, such as NASA's Orbiting Carbon Observatory-2

(OCO-2) and OCO-3, the TROPOspheric Monitoring Instrument (TROPOMI) onboard the Copernicus Sentinel-5 Precursor (S5P) satellite (Veefkind et al., 2012), MethaneSat for methane, and a planned constellation of GHG monitoring satellites (e.g., GOSAT-GW).

The increased volume of observational data sets provide more constraints to estimate GHG emissions. However, current methods do not scale well with the increasing number of observations. Inferring GHG emissions using atmospheric observations is conventionally done via atmospheric flux inversions (e.g., Jiang et al., 2017; White et al., 2019; Turner et al., 2020). The state of the art in atmospheric flux inversions relies on either Eulerian models or Lagrangian particle dispersion models (LPDMs) to simulate atmospheric transport, which provides the means of relating observations to surface fluxes. For example, the four-dimensional variational (4D-Var) method uses the adjoint of Eulerian models to calculate sensitivities of GHG concentrations to surface fluxes (Baker et al., 2006; Henze et al., 2007; Jiang et al., 2017; Qu et al., 2022). Kalman filters are also widely used in flux inversions, which calculate covariance matrices between prior fluxes and GHG concentrations simulated by Eulerian models to estimate posterior fluxes (Feng et al., 2009; Kang et al., 2011; Miyazaki et al., 2017, 2020). Alternatively, LPDMs can be used to calculate the sensitivity of each observation to its upwind sources by simulating the trajectories of an ensemble of particles advected backward in time (Lin et al., 2004; Fasoli et al., 2018; Jones et al., 2007b; Pisso et al., 2019). The sensitivity of each receptor to its upwind sources, termed as the receptor's "footprint", can then be used to estimate fluxes inversely (e.g., Stohl et al., 2003, 2009; Jones et al., 2007a; Lin et al., 2004, 2021; Stein et al., 2015; Turner et al., 2020). These methods based on full-physics models are becoming prohibitively expensive due to the large computational burden of running high-resolution atmospheric transport models for dense observing systems. The 4D-Var method runs the forward and adjoint models iteratively to optimize the a posteriori emission, which is hard to parallelize. Kalman filters could benefit from parallelism, however, they still require the forward model and the computational cost scales up with the number of processors used (e.g., Houtekamer and Mitchell, 2001).

Here we present a machine learning-based emulator, FootNet, to efficiently calculate footprints of ground-based receptors with a high fidelity at kilometer-scale spatial resolution. The footprint emulator reduces the computational and storage cost of Lagrangian model-based flux inversion systems by 2–3 orders of magnitude, which will better accommodate the increased volume of GHG observations. We show the evaluation of the performance of FootNet using independent data sets. Finally, we assess the relative importance of the input variables of FootNet using the permute-and-prediction (PaP) method.

## 2 Construction of the FootNet emulator

Training of the FootNet model is a supervised learning process, which requires ground truth to guide the optimization of the model parameters. Here, we use a full-physics model to generate the ground truth. We simulate footprints using the Stochastic Time-Inverted Lagrangian Transport (STILT) model (Lin et al., 2003; Fasoli et al., 2018), a Lagrangian particle dispersion model (LPDM). STILT simulations are conducted for two regions: the Barnett Shale region in Texas, and the SF Bay Area in California (see Figure 2). These two regions are chosen because one has simple topography (the Barnett Shale) whereas the other is topographically complex (SF Bay Area). As such, these regions represent limiting cases for the construction and

evaluation of the emulator. Further, the combination of two regions will help prevent from overfitting the model to a single location. For the SF Bay Area, STILT simulations are run from 2018 to 2020 with receptors located at realistic sites deployed in the BEACO$_2$N network (see http://beacon.berkeley.edu and Shusterman et al. (2016)). Footprints for the Barnett Shale region are generated from a 1-week WRF-STILT simulation in 2013 (Turner et al., 2018). All STILT runs are conducted within $400 \times 400$ km$^2$ domains at $1 \times 1$ km$^2$ spatial resolution (see Figure 1). The footprints are integrated 72 hours backwards from the measurement time, because of the 400 km $\times$ 400 km domain used by the FootNet model. The time integration period could change depending on the spatial and time scales of inversion systems.

The output of FootNet is a source-receptor relationship (i.e., footprint, **H**), which represents the sensitivity of atmospheric concentrations at a receptor site to emissions upwind of the receptor. This relationship between the measured concentrations and the emissions in the upwind area can be formulated as

$$\mathbf{y} = \mathbf{H}\mathbf{x} + \mathbf{b}$$

where **y** represents the measured concentration, **x** is the emission fluxes in a domain around the measurement location, and **b** is the background concentration upwind of the domain. The units of **y** and **x** can be expressed as dry air mixing ratio (ppb) and flux rates (nmol m$^{-2}$ s$^{-1}$), respectively (Lin et al., 2003). The source-receptor relationships, $\mathbf{H} = \partial \mathbf{y} / \partial \mathbf{x}$, therefore have units of ppb / (nmol m$^{-2}$ s$^{-1}$).

The calculation of measurement footprints is independent of the observed gas concentrations and could be constructed using meteorological variables only. As shown in Table 1, we use 4 physical parameters from the NOAA High-Resolution Rapid Refresh (HRRR; Benjamin et al. (2016)) model as the input variables, including the 10-meter zonal wind speed (U10M), 10-meter meridional wind speed (V10M), planetary boundary layer height (PBLH) and surface pressure (PRSS). The FootNet model receives input variables at the measurement time ($t_0$) and 6 hours before the measurement time ($t_0$-6h) to predict footprints at $t_0$. The choice of 6 hours backwards was determined by a series of sensitivity tests on the amount of history information in the input data (see Supplemental Section S1). We found that including history information from more than 6 hours could not further improve the performance of FootNet in the emulation (see Figures S1-3). However, we note that the results from the sensitivity tests could depend on the spatial and temporal scale and resolution of the specific inversion problems. Evaluation of the necessary history information in other spatio-temporal regions is warranted.

We scale the input variables to a similar magnitude for the stabilization of the training process (see Table 1). The output of FootNet is measurement footprints and is transformed by the natural logarithm function to reduce the skewness of the distribution of footprint values. The transformed footprints are filtered to remove values smaller than -20 and then shifted by +20, corresponding to a scaling of the raw footprints by $e^{20}$. We find that including Gaussian plumes (see Figure 1) as one of the input variables could significantly improve the performance of FootNet. The Gaussian plumes are calculated using the Gaussian plume model (e.g., Stern, 1976; Dobbins, 1979; Zannetti, 1990, among others) with reversed wind fields starting from the measurement site, which are used as the initial guess of the upwind areas and the measurement footprints. The Gaussian plumes can be efficiently calculated as a Hadamard product from inputs listed above and, as such, adds minimal computational expense. The Gaussian plumes also provide a localization for FootNet in that it contains the information about measurement

location and provides an initial guess for the spatial structure of the footprint. The FootNet model is trained to learn the nonlinear transformation from the idealized Gaussian plumes to measurement footprints using the meteorological fields. The input variables are interpolated to the $400\times400$ km$^2$ domain and the $1\times1$ km$^2$ spatial resolution of footprints.

The model structure underlying the footprint emulator is the U-net model (Ronneberger et al., 2015), which is now broadly applied in the field of Earth Science (Ghorbanzadeh et al., 2021; He et al., 2022a, b; Zemskova et al., 2022; Tucker et al., 2023; He et al., 2024). A schematic diagram of the model architecture is shown in Figure 1. The model consists of 4 convolutional blocks and 4 up-convolutional blocks. Each convolutional block is a sequence of two convolutional layers with $3 \times 3$ kernels and one $2 \times 2$ max-pooling layer. In each convolutional layer, the input images will be performed the convolution calculation

with $3 \times 3$ kernels that will scan the whole images to generate output images. In max-pooling layers, the input images will be down-sampled by taking maximum values in each $2 \times 2$ region in the images. Similarly, each up-convolutional layer has one $2 \times 2$ up-convolutional layer followed by two $3 \times 3$ convolutional layers. Up-convolutional layers perform transposed convolution operation with $2 \times 2$ kernels scanning input images. The outputs from convolutional layers are all transformed by the Rectified Linear Unit (ReLU) function to increase non-linearity in predictions. In the training process, the entries of $3 \times 3$

convolutional kernels and $2 \times 2$ up-convolutional kernels will be optimized along the partial gradients of a loss function that measures the difference between the truth and FootNet predictions. More details about deep learning architectures could be found in Goodfellow et al. (2016).

     We train and evaluate the emulator using a data set with 10,000 natural log-transformed footprints (logH) from the Barnett Shale and 10,000 footprints from the SF Bay Area as the truth. We apply natural logarithm transformation to the measurement

footprints because their values are often highly skewed, which could be challenging for the FootNet model to learn in the training process. The combined data set is randomly split to 85% as the training data set and 15% as the test data set. The test data set is independent of the training process. 15% of the training data set is used as a validation data set during the training process to prevent overfitting. We use mean squared error as the loss function and the Adam optimization algorithm.

     We use the Intersection over Union (IoU) to measure the accuracy of the area of footprints predicted by FootNet, which is

defined as follows:

$$IoU = \frac{|Y \cap \hat{Y}|}{|Y \cup \hat{Y}|} \tag{1}$$

Here, $Y$ and $\hat{Y}$ stand for the ground truth (footprints simulated by STILT) and the FootNet predictions, respectively. The absolute value bars ($|\cdot|$) here refer to the area of a region. Specifically, the intersection, $|Y \cap \hat{Y}|$, calculates the area of the region where both the truth and FootNet predictions show non-zero footprints. Similarly, the union, $|Y \cup \hat{Y}|$, represents the

area of the region where either the truth or FootNet predictions show non-zero footprints. IoU is widely used to evaluate the ability of deep learning models to make accurately localized predictions. We also compute Pearson correlation coefficients ($r$) for footprints in the intersection areas between the truth, as simulated by STILT, and the corresponding FootNet predictions to help assess the performance.

     Ultimately, we are interested in better understanding what drives the predictions from the FootNet model. As such, we use

the permute-and-prediction (PaP) method to calculate the importance of input variables for footprint emulation, which provides

some interpretability of the FootNet model (Fisher et al., 2019). The PaP method estimates variable importance by permuting each input variable with different data samples, and the subsequent performance change represents FootNet's sensitivity to the permuted variable. We estimate variable importance by calculating performance changes in correlation, the IoU, and the root mean square error (RMSE) of the predicted footprints.

## 3   Evaluating performance of the FootNet emulator

Figure 3 demonstrates the evolution of FootNet predictions during the training process and the overall performance of FootNet after the training converges. Figure 3D shows a footprint simulated by the STILT model from the test data set, where the footprint is highly nonlinear with a change in direction near the receptor. The corresponding FootNet predictions are shown in Figures 3 (A-C). After iteration A (shortly after the training starts), the FootNet predicts measurement footprints around the receptor with a large negative bias and low correlation coefficient of 0.49. Iteration B is about halfway of the training process, after which the FootNet prediction better captures the general shape of the footprint and the correlation is improved to 0.61. The training stops after iteration C. The final FootNet prediction has enriched details and attains a correlation coefficient of 0.75. Compared to the truth in Figure 3D, the IoU of FootNet predictions improves from 0.28 after iteration A to 0.51 after iteration B, and attains a final IoU of 0.76 (see Figure 3E). Figure 3F shows the comparison between the truth and FootNet predictions for all footprints in the test data set. FootNet predictions show a slight negative bias compared to footprints simulated using the full-physics STILT model. The overall Pearson correlation coefficient ($r$) between FootNet predictions and STILT simulations is 0.58. We conclude that FootNet is able to emulate the source-receptor relationship in both simple (Barnett Shale, TX) and complex (SF Bay Area, CA) meteorological conditions with high fidelity. However, it is worth mentioning here that we find some performance degradation using an alternative splitting of the data based on different time periods. Because the training data set used to construct version 1 of FootNet has a relatively small size, similarities between samples are hard to fully avoid by randomly selecting training data samples, which could lead to generalizability issues when using FootNet version 1 over regions and time periods too different from the training data set. This generalizability issue could be largely mitigated by increasing the volume of the training data set in the future (Dadheech et al., 2024).

We then evaluate the performance of FootNet in predicting individual footprints for the two regions. Figure 4 shows footprints from STILT and FootNet for the two regions: the Barnett Shale and the SF Bay Area. Figures 4A and 4E show results from the simple case (Barnett Shale, TX), where the footprint is similar to an idealized Gaussian plume with time-reversed winds. FootNet well captures both the magnitudes and spatial patterns of the footprint, with an IoU of 0.73 and a correlation coefficient of 0.54. Figures 4B and 4F demonstrate a more complicated meteorological scenario in the Barnett Shale region. The IoU metric and correlation coefficient between the STILT footprint and the FootNet prediction are 0.71 and 0.61, respectively, for this more complex scenario.

Atmospheric transport in the SF Bay Area is decisively more complex because the region includes steep topography, air-sea interactions, and numerous valleys and deltas. Figures 4C and 4D show results from the full-physics model for the SF Bay Area. Emulation of footprints in the Bay Area is more challenging and with an overall degraded fidelity as compared to the

Barnett Shale region. Figures 4C and 4G show a receptor with the bulk of the footprint in the Northwest quadrant of the domain, as a result of typical summertime meteorology in the SF Bay Area with westerly flow bringing air masses into the SF Bay Area past the Golden Gate Bridge. The shape and the magnitude of the footprint is predicted by FootNet with an IoU of 0.53 and the correlation coefficient to be 0.83. Figures 4D and 4H show a more complex meteorological scenario, where the FootNet prediction has an IoU of 0.56 and the correlation is 0.78 as compared to STILT.

There have been other methods developed to improve the efficiency of footprint calculations. For example, Roten et al. (2021) uses nonlinear weighted averaging to interpolate footprints from locations near the receptors. Fillola et al. (2023) develops a similar footprint emulator based on gradient-boosted regression trees (GBRTs), at a coarse spatial resolution (20–30 km in mid-latitudes) and 10 grid cells around the measurement location. Compared to previous work, the FootNet model reproduces the full-physics model with high fidelity at high-resolution. This is remarkable given the complex topography and meteorology of the regions studied here could complicate transport at kilometer scale and the emulation of footprints. Moreover, FootNet only takes meteorological fields and the idealized Gaussian plume as its input. No additional LPDM simulations are needed to generate footprint predictions after the training process.

Emulation of footprints using the FootNet model brings co-benefits for computational efficiency and storage cost, and better facilitates the application of LPDM-based flux inversion systems with dense observing systems. To conduct kilometer-scale emission inversions using one day of observations made at the 40 BEACO$_2$N sites in the SF Bay Area (approx. 650 observations per day), it takes the full-physics STILT model about 640 core-hours to generate the required footprints. The generation of each footprint prediction takes ∼1 s on a 32-core compute node, which can be further reduced to 0.08 s on an NVIDIA A2 graphics processing unit (GPU). Only 6 minutes are required for FootNet on an A2 GPU node to generate the required footprints for one day of BEACO$_2$N measurements. The storage requirement also makes it impractical to use full-physics models in high-resolution flux inversions with dense observations. Hourly footprints for one week of BEACO$_2$N measurements would require 4-terabyte storage space for future re-use. With FootNet, footprints could be generated in near real-time and there is no need to store the computed footprints.

Figure 5 shows the ranking of variable importance for FootNet calculated using the PaP method on 1000 randomly selected data samples. Overall, the most important meteorological variables are the 10-meter wind speeds, which lead to a 0.2∼0.3 decrease in correlation and the IoU drops by 0.1∼0.2 after being permuted. Permuting Gaussian plumes degrades the correlation and the IoU of FootNet predictions by 0.1 and 0.03, respectively. We find less sensitivity of FootNet predictions to surface pressure and planetary boundary layer height than other input variables. This is because we only have training data from two locations in the current version of the model, and these two meteorological fields show much less variability than the wind fields in the training data set. We still include surface pressure and PBL height as input variables because they are essential information for the generation of measurement footprints. We expect to see greater importance for surface pressure and PBL height for a general version of FootNet trained using footprints from more locations in the future. Figure 5 also shows that input variables from $t_0$-6h have consistently greater importance than $t_0$.

The PaP method only provides a rough estimate of variable importance, and the inter-correlation between input variables can lead to an inflation of the feature importance (Hooker et al., 2021). Nevertheless, the estimated variable importance for

FootNet is in alignment with with our understanding about the calculation of footprints in a full-physics model, which relies
on the advection of particles driven by precomputed wind fields. The Gaussian plume is also identified as highly important,
because it is the only input field providing information about the location of receptors.

## 4 Conclusions

We described the development of a machine learning-based emulator of surface measurement footprints, FootNet. The footprint
emulator can be used to improve the computational efficiency of estimating high-resolution GHG fluxes using measurements
made by dense observing systems. The FootNet model was trained and evaluated using footprints simulated by the STILT
full-physics model for the SF Bay Area and the Barnett Shale region. We showed the convergence of FootNet predictions to
the STILT truth as the training iterates. The overall correlation between FootNet predictions and the STILT truth in the test data
set reaches 0.58 after full convergence. The emulator well predicts both the extents and magnitudes of footprints with a high
fidelity. We estimated importance of input variables using the PaP method to improve the interpretability of the FootNet model.
We found 10-meter wind speeds and Gaussian plumes have the greatest importance for the emulation of footprints. Emulation
of footprints using FootNet brings co-benefits for computationally efficient and reducing storage cost, which makes it feasible
to deliver high-resolution estimates of GHG fluxes in near real-time using proliferated dense observing systems in the future.

Due to the computational cost required by the generation of high-resolution footprints, we only included footprints generated
from previous studies for the two locations in training version 1.0 of FootNet. We are actively generating new footprints at 1
km from a broader region to further improve the emulator's performance, especially in regions with different meteorological
conditions from the two locations used in this study (Dadheech et al., 2024). Generalizing this source-receptor emulator to
other regions is being tackled in the next version of FootNet.

*Code and data availability.* We use the full-physics Stochastic Time-Inverted Lagrangian Transport Model (STILT) to simulation footprints
for the training of FootNet. The STILT model could be accessed from https://uataq.github.io/stilt/ (Fasoli et al., 2018). Footprints simulated
by the STILT model are available through Turner et al. (2018) and Turner et al. (2020). Examples of the footprints used in the train-
ing process could be downloaded from https://zenodo.org/records/12803617, https://zenodo.org/records/12803736, and https://zenodo.org/
records/12803855. The meteorological variables are from the High-Resolution Rapid Refresh (HRRR) data product, which is available at
https://rapidrefresh.noaa.gov/hrrr/ (Dowell et al., 2022; James et al., 2022). The repository of the code used in the manuscript is publicly
available at https://zenodo.org/records/12752655.

*Author contributions.* T.L.H., N.D., and A.J.T. designed the research study; T.L.H. and N.D. built and trained the model; T.L.H. and N.D.
performed research and analyzed results; T.L.H., N.D., T.M.T. and A.J.T. contributed to revising and editing the manuscript.

*Competing interests.* The authors declare no conflict of interest.

*Acknowledgements.* This work is supported by a NASA Early Career Faculty Grant (80NSSC21K1808) to A.J.T. and NASA FINESST Grant (80NSSC22K1557) to N.D. We acknowledge funding from Environmental Defense Fund, whose work is supported by gifts from Signe Ostby, Scott Cook and Valhalla Foundation. This research received support through Schmidt Sciences, LLC.

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

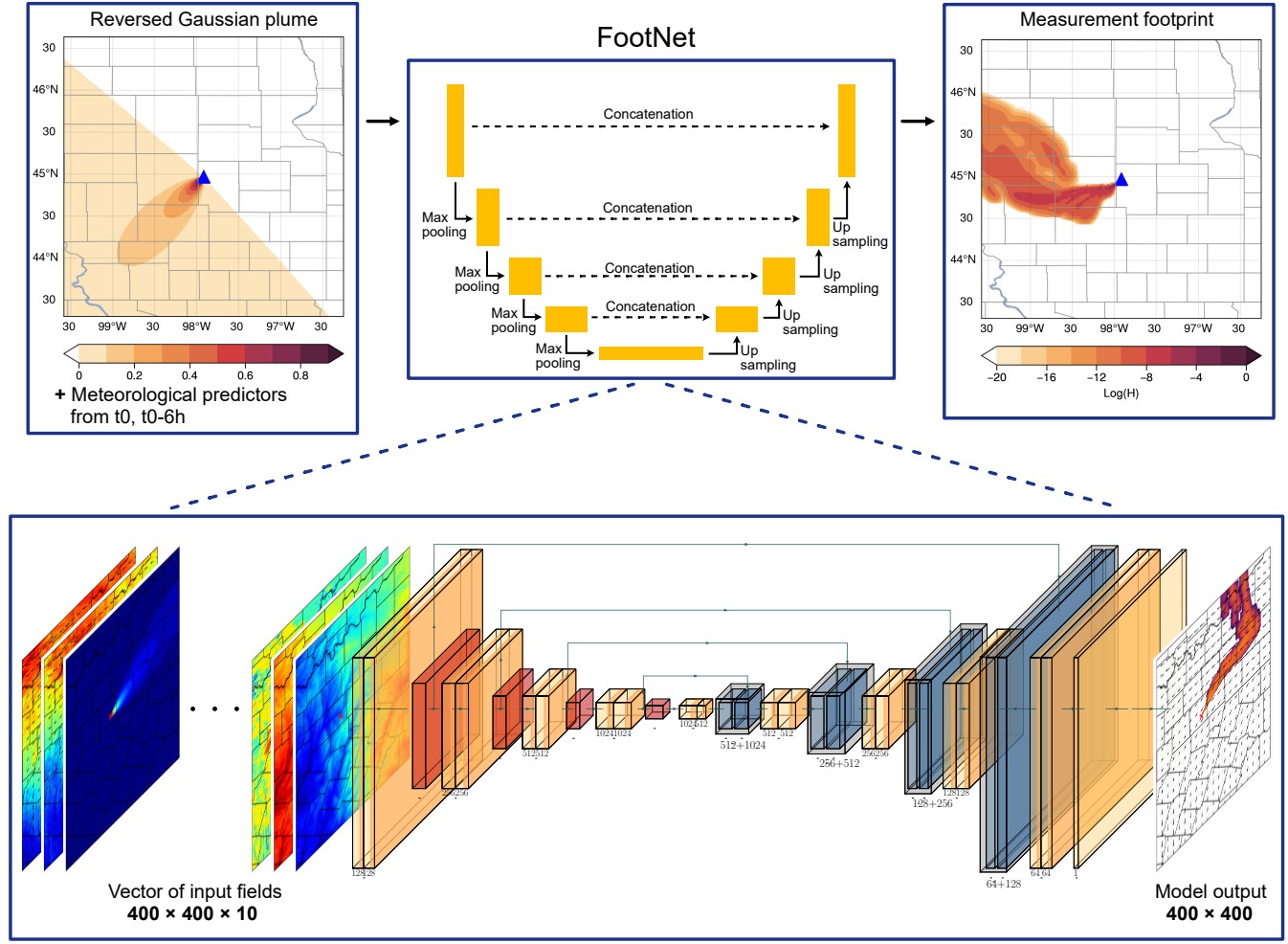

**Figure 1.** Top row shows the schematic diagram of the FootNet model. Detailed structure of FootNet is shown at the bottom. The orange boxes indicate 3×3 convolutional layers. The red boxes represent 2×2 max-pooling layers. The light blue boxes are 2×2 transposed convolutional layers. The dark blue boxes represent the latent vectors concatenated from previous layers (shown as parallel arrows on top).

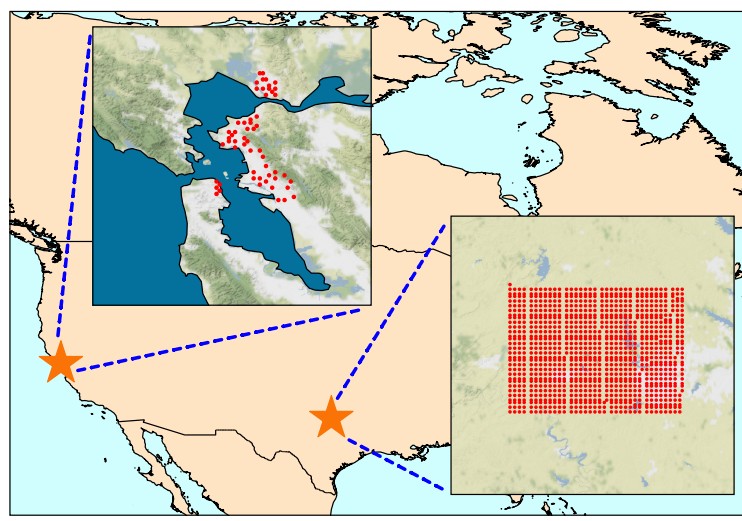

**Figure 2.** Location of receptors for simulations of measurement footprints using the STILT model. Receptors in the SF Bay Area are located at sites in the BEACO₂N network (Shusterman et al., 2016). Receptors in the Barnett Shale region are at locations used in Turner et al. (2018). Map tiles are from © Stamen Design, under a Creative Commons Attribution (CC BY 3.0) license.

**Table 1.** Information about input variables of FootNet.

| Variable (Unit) | Description | Time steps | Scaling factor |
|---|---|---|---|
| Gaussian plume | Idealized plumes calculated using reversed winds | $t_0$, $t_0$-6h | 1 |
| U10M (m/s) | 10-meter U-component of wind | $t_0$, $t_0$-6h | 10 |
| V10M (m/s) | 10-meter V-component of wind | $t_0$, $t_0$-6h | 10 |
| PBLH (m) | PBL height | $t_0$, $t_0$-6h | 1e-3 |
| PRSS (hPa) | Surface pressure | $t_0$, $t_0$-6h | 1e-3 |

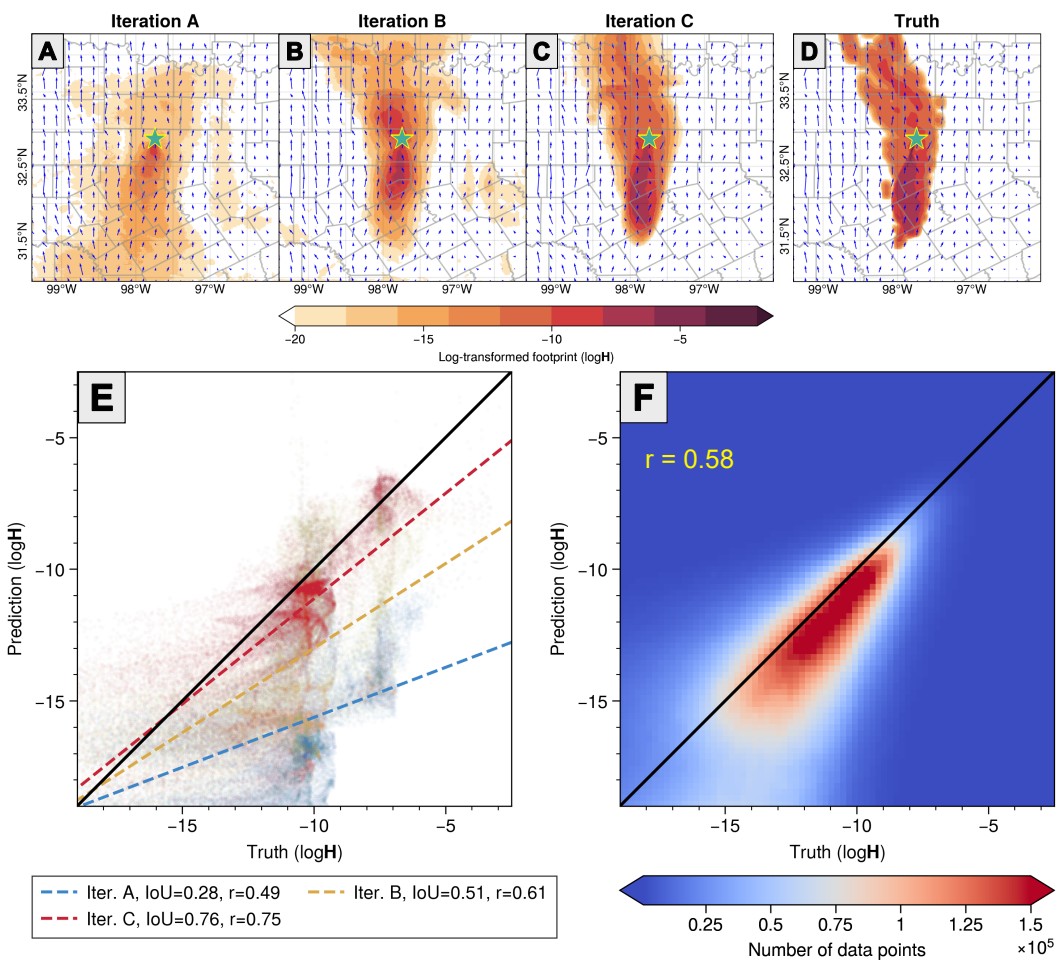

**Figure 3.** Convergence of the training process and evaluation of the model performance on the independent test data set. Footprints are transformed using natural logarithm. The unit of footprints is ppb / (nmol $m^{-2}$ $s^{-1}$) before the transformation. (A-C) FootNet predictions from three stages in the training process, corresponding to the truth in (D). The blue arrows represent wind vectors, and the green stars show the location of the receptors. (E) Comparison between footprints simulated by STILT and FootNet predictions in (A-C). (F) Two-dimensional histogram of all natural log-transformed footprint (logH) values simulated by STILT and the corresponding predictions made by FootNet from the test data set.

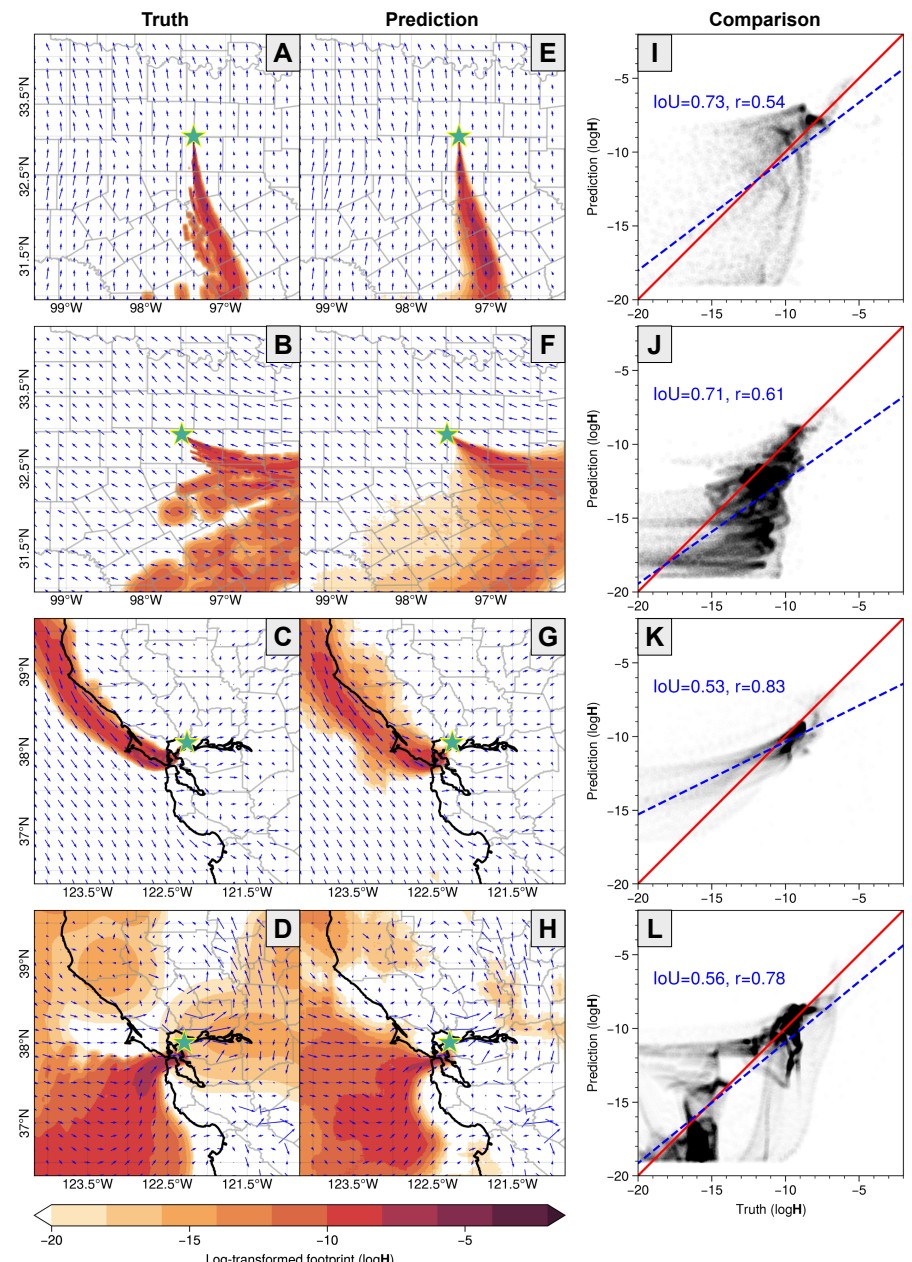

**Figure 4.** Evaluation of individual FootNet predictions from the test data set. Footprints are transformed using natural logarithm. The unit of footprints is ppb / (nmol m$^{-2}$ s$^{-1}$) before the transformation. (A-D) Footprints simulated by the full-physics STILT model for the Barnett Shale region and the SF Bay Area. (E-H) Footprint predictions made by FootNet corresponding to (A-D). The blue arrows represent wind vectors, and the green stars show the location of the receptors. (I-L) Comparison and correlation between the truth and predictions for the four examples.

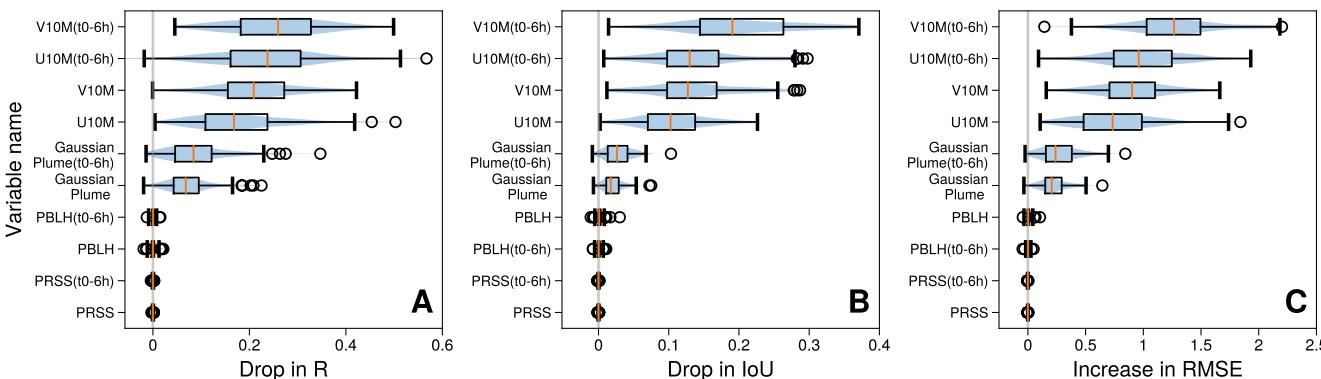

**Figure 5.** Rankings of variable importance estimated using the permute-and-predict (PaP) method on 1000 data samples. (A-C) Variable importance shown as drop in correlation, drop in the IoU, and increase in the RMSE after permuting the 10 input variables. Orange lines show the medians. Boxes indicate ranges from the first quartiles to the third quartiles. Whiskers are the 1.5 interquartile ranges (IQRs) from the boxes. Circles are outliers.