# Peer review of "FootNet v1.0: Development of a machine learning emulator of atmospheric transport"

_EGUsphere, 2024_

## Author Comment (AC3)

We thank the reviewers for raising important questions and offering suggestions, which have helped us improve our manuscript. We made several edits to the manuscript to address the reviewers' comments. Our response to the reviewers is color-coded in blue.

**Reviewer #1:**

The article mentions that the four-dimensional variational method (4D-Var) and the Kalman filter method (both of which are based on full physical field models for flux inversion) become expensive with increased resolution. However, the article does not clearly demonstrate the advantages of the deep learning-based "footprint" simulator compared to these full physical field model-based methods. The author provided some time consumption data, but they are not directly comparable since 640 core-hours and 32 cores for 1 second cannot be directly compared without knowing whether both can achieve parallelism and the efficiency of that parallelism.

The motivation for this work is to accelerate GHG flux inversions.  As the reviewer mentions, we provide background on various approaches for GHG flux inversions.  With an LPDM-based inversion, the first step is to construct source-receptor relationships (footprints).  The scope of this paper is to construct those footprints.  This manuscript does not actually conduct GHG flux inversions but we have ongoing work (Dadheech et al., submitted) using this ML surrogate model in a GHG flux inversion.  As such, we do not have directly comparable numbers to the 4D-Var or Kalman filter methods in this manuscript.  The focus of this section was to highlight, at broad scales, where these methods can be accelerated and where they cannot.  We elaborate on this in the manuscript.

Line 54–57: "The 4D-Var method runs the forward and adjoint models iteratively to optimize the a posteriori emission, which is hard to parallelize. Kalman filters could benefit from parallelism, however, they still require the forward model and the computational cost scales up with the number of processors used (Houtekamer and Mitchell, 2001)."

References:

Houtekamer, P. L., and H. L. Mitchell, 2001: A Sequential Ensemble Kalman Filter for Atmospheric Data Assimilation. Mon. Wea. Rev., 129, 123–137, https://doi.org/10.1175/1520-0493(2001)129<0123:ASEKFF>2.0.CO;2.

Additionally, the author did not theoretically analyze why the deep learning-based "footprint" simulator can reduce costs, such as whether the speedup is due to the

structure of the machine learning algorithm or hardware acceleration by GPUs. If both contribute, what is the respective contribution of each?

Thank you for this comment. We mentioned in the text about the speedup contributed by the model structure and the use of GPUs. We have reformatted the paragraph to make it clear that the performance is largely contributed to by the machine learning algorithm.

Line 166–175: "Emulation of footprints using the FootNet model brings co-benefits for computational efficiency and storage cost, and better facilitates the application of LPDM-based flux inversion systems with dense observing systems. To conduct kilometer-scale emission inversions using one day of observations made at the 40 BEACO2N sites in the SF Bay Area (approx. 650 observations per day), it takes the full-physics STILT model about 640 core-hours to generate the required footprints. The generation of each footprint prediction takes ~1 s on a 32-core compute node, which can be further reduced to 0.08 s on an NVIDIA A2 graphics processing unit (GPU). Only 6 minutes are required for FootNet on an A2 GPU node to generate the required footprints for one day of BEACO2N measurements. The storage requirement also makes it impractical to use full-physics models in high-resolution flux inversions with dense observations. Hourly footprints for one week of BEACO2N measurements would require 4-terabyte storage space for future re-use. With FootNet, footprints could be generated in near real-time and there is no need to store the computed footprints."

The author's description of the input and output data metrics is unclear. For instance, why is the most important metric, observed concentration data, not included as an input? Shouldn't the output data be a spatial distribution of emissions? Why is there only a logH? What does H represent?

As mentioned above, the goal of this manuscript is to construct source-receptor relationships (i.e., footprints). For non-reactive tracers, the footprint is independent of the observed concentration. We have modified the text to clarify this:

Line 77–79: "The output of FootNet is source-receptor relationships (i.e., footprints, H), which represent the sensitivity of concentration measurements to emissions in the upwind area of the measurement site (with units like ppb/(nmol m^-2 s^-1)). The calculation of measurement footprints is independent of the observed gas concentrations and could be constructed using meteorological variables only."

Line 111–112: "… We apply log transformation to the measurement footprints because their values are often highly skewed, which could be challenging for the FootNet model to learn in the training process. …"

What is the Gaussian plume, and what is its physical significance and calculation process? Its inclusion or exclusion's impact on the results is not shown in Figure 2.

Gaussian plumes are calculated using a Gaussian plume dispersion model (e.g., Stern, 1976; Dobbins, 1979; Zannetti, 1990, among others) with reversed wind fields, which are used as the initial guess for measurement footprints. We did not include the results without Gaussian plumes, because, as shown in Figure 5, Gaussian plumes and wind fields are the most important input predictors for footprint emulations and excluding them will lead to significantly degraded performance.

Line 90–92: "… The Gaussian plumes are calculated using the Gaussian plume model (e.g., Stern, 1976; Dobbins, 1979; Zannetti, 1990, among others) with reversed wind fields starting from the measurement site, which are used as the initial guess of the upwind areas and the measurement footprints. ..."

Why is data only needed 6 hours in advance and not earlier? Has the author conducted similar tests with data from earlier times?

Great question.  We conducted a series of sensitivity tests on the amount of history information in the input. The corresponding results and the comparisons against the version shown in the main text are now presented in the Supplementary Materials. We have included a discussion on the amount of history information in the main text.

Line 83–88: "… The choice of 6 hours backwards was determined by a series of sensitivity tests on the amount of history information in the input data (see Supplemental Section S1). We found that including history information from more than 6 hours could not further improve the performance of FootNet in the emulation (see Figures S1-3). However, we note that the results from the sensitivity tests could depend on the spatial and temporal scale and resolution of the specific inversion problems.  Evaluation of the necessary history information in other spatio-temporal regions is warranted."

The consistency of the results in this study is not well-established, and some statistical indicators are not very high. The author should provide a detailed analysis of the reasons for the lack of accuracy and suggest directions for future improvement.

In contrast to the reviewer, we find the performance to be quite good.  There is substantial uncertainty in atmospheric transport.  Using two different LPDMs (e.g., STILT and FLEXPART) to simulate the same source-receptor relationship would lead to larger disagreement than observed here.  We have ongoing work to evaluate the

performance of FootNet in different contexts. Most relevant, we have used FootNet as a surrogate for the full-physics model in a GHG flux inversion (Dadheech et al., submitted). We elaborate on additional methods to improve the ML model going forward in the updated manuscript:

Line 204–208: "Due to the computational cost required by the generation of high-resolution footprints, we only included footprints generated from previous studies for the two locations in training version 1.0 of FootNet. We are actively generating new footprints at 1~km from a broader region to further improve the emulator's performance, especially in more general use cases with different meteorological conditions from the two locations used in this study (Dadheech et al., submitted)."

**Reviewer #2:**

I think the article would benefit from much more extensive evaluation of the emulated footprints. The figures in the manuscript examine the correlation between the log of the true footprints against the log of the emulated footprints. At the end of the day, many modelers ultimately care about the accuracy of simulated greenhouse gas or air pollution mixing ratios. Hence, I personally think that the correlation between the footprint values is necessary but not sufficient to convince many modelers (including myself) to use a tool like FootNet. For example, let's suppose one used these footprints to model CO2 and CH4 mixing ratios. Would these footprints capture peaks and troughs in CO2 and CH4 (I.e., suppose one were to plot CO2 and CH4 at individual sites as timeseries.)? Would the emulated footprints capture spatial variability in atmospheric CO2 or CH4 levels? Suppose there were CH4 super-emitters scattered in the Barnett Shale. Would the emulated footprints accurately capture the impact of those super-emitters on downwind atmospheric observations? Let's say one were to model CO2 using these emulated footprints. Would those footprints capture diurnal variability in CO2 mixing ratios (i.e., due to variability in both fluxes and boundary layer dynamics)?

This is a fantastic point. We allude to this in response to Reviewer #1, but the scope of this paper is on the construction of this emulator in two limiting cases: 1) the Barnett Shale because it is an "easy" region and 2) the SF Bay Area because it is a "complex" region. We have ongoing work to evaluate the performance of the emulator in a GHG flux inversion that we recently submitted to ACP (Dadheech et al., submitted). There was extensive work required to make this directly usable in a GHG flux inversion and some counterintuitive results. Therefore, separating the first demonstration (this paper) from the incorporation into a GHG flux inversion framework seemed warranted. Additionally, this paper was initially submitted to GRL and, as such, is work that was conducted before the inversion work. We view this work as a demonstration that we

can sufficiently emulate these source-receptor relationships. We have now included this in the conclusions section.

Line 204–208: "Due to the computational cost required by the generation of high-resolution footprints, we only included footprints generated from previous studies for the two locations in training version 1.0 of FootNet. We are actively generating new footprints at 1~km from a broader region to further improve the emulator's performance, especially in regions with different meteorological conditions from the two locations used in this study (Dadheech et al., submitted). With the next version of FootNet trained for more general use cases, the performance of FootNet in an inversion system could be further assessed in the future."

In addition, I imagine it might not always be practical to use 85% of data for training. For example, if one wanted to run footprints for a large satellite dataset, it might (hypothetically) only be feasible to use 5% or 10% of the data for training. In that case, one would need to train FootNet on a limited number of data points and then run the trained FootNet algorithm on a much larger number of data points. Do you have a sense of how FootNet would perform in this circumstance?

We think there may be some confusion here.  Once trained, there would be no need to run more training data.  Our ultimate goal is to build a generalizable ML model that can be used for surface or satellite data over any region.  This manuscript is a first step showing that the ML model performs well in two limiting cases. The second step (Dadheech et al., submitted) is demonstrating how to use this in a GHG flux inversion. The third step is to generalize this model to work for any spatio-temporal location over the CONUS (Dadheech et al., in prep).  Again, once trained, the users would not need to make additional training data.

Line 204–208: "Due to the computational cost required by the generation of high-resolution footprints, we only included footprints generated from previous studies for the two locations in training version 1.0 of FootNet. We are actively generating new footprints at 1~km from a broader region to further improve the emulator's performance, especially in regions with different meteorological conditions from the two locations used in this study. …"

Both of the case studies described in this paper are for small geographic regions (e.g., San Francisco and the Barnett Shale). Let's say one wanted to use FootNet across the entire US or across the entire globe. For these larger spatial scales, I imagine there are more variable and diverse transport patterns in different regions. In this circumstance, one would want FootNet to capture all those transport patterns in different regions. By

contrast, San Francisco and the Barnett Shale, by factor of their limited geographic size, might have a more limited set of transport patterns to capture. These different circumstances might necessitate very different approaches to the training data, and the resulting emulated footprints might not have the same fidelity or accuracy.

Fantastic question, this is exactly what we are working towards: Dadheech et al. (in prep).

**Specific suggestions:**

- It would be helpful to include line numbers on future versions of the manuscript. Doing so would make it easier to discuss specific lines of the manuscript.

  Apologies, line numbers are now included in the revised manuscript.

- Abstract: What does "near-real-time" mean in this context?

  The term "near real-time" means the generation of measurement footprints could be generated using FootNet on order of milliseconds, which could be immediately used for emission inversion.

- Intro: "The sensitivity of each receptor to its upwind sources, termed as the receptor's "footprint", can then be used to estimate fluxes inversely (e.g., Turner et al., 2020)." There are a bunch of other good references that could be used as examples here going back to the early 2000s.

  Thank you for the suggestion. We have added more references to the discussion.

- Intro: "The footprints are integrated 72 hours backwards from the measurement time." A lot of continental-scale studies using STILT use footprints that are integrated 10 days backward from the measurement site. Do you think the approach developed here is applicable to those longer time scales?

  Thank you for the comment. The footprints were integrated 72 hours backwards because of the 400 km x 400 km domain used by the FootNet model. We acknowledge that the integration period could change depending on the spatial and time scales. This discussion is now added to the text.

  Line 74–76: "The footprints are integrated 72 hours backwards from the measurement time, because of the 400 km x 400 km domain used by the FootNet model. The time integration period could change depending on the spatial and time scales of inversion systems."

- Sect. 2: "Each convolutional block includes two convolutional layers with 3 × 3 convolutional kernels and one 2 × 2 max-pooling layer." What is a convolutional block, convolutional kernel, and max-pooling layer? I suspect that most readers won't be familiar with these terms.

  We have added some definition and explanation of the terms, and cited references to help readers.

  Line 102–111: "… Each convolutional block is a sequence of two convolutional layers with 3x3 kernels and one 2x2 max-pooling layer. In each convolutional layer, the input images will be performed the convolution calculation with the 3x3 kernels and the kernels will scan the whole images to generate the output images. In max-pooling layers, the input images will be down-sampled by taking maximum values in each 2x2 region in the images. Similarly, each up-convolutional layer has one 2x2 up-convolutional layer followed by two 3x3 convolutional layers. Up-convolutional layers perform the transposed convolution operation with 2x2 kernels scanning input images. The outputs from convolutional layers are all transformed by the Rectified Linear Unit (ReLU) function to increase non-linearity in predictions. In the training process, the entries of 3x3 convolutional kernels and 2x2 up-convolutional kernels will be optimized along the partial gradients of a loss function, which measures the difference between the truth and FootNet predictions. More details could be found in Goodfellow et al. (2016)."

- Sect. 3 "The overall correlation between FootNet predictions and STILT simulations..." Does this line refer to r or r^2?

  The overall correlation refers to the Pearson correlation (r). We have modified the sentence to make it clear.

  Line 142–143: "The overall Pearson correlation coefficient (r) between FootNet predictions and STILT simulations is 0.58."

- Pg. 8, line 1: This paragraph feels like it could use a better topic sentence. You've just finished describing the training process for a footprint calculation in the Barnett Shale. What topic or concept are you going to describe next? I think the answer to this question would better guide the reader and give the reader a better idea of what to anticipate.

  Thank you for the suggestion. We have rewritten the topic sentence for the paragraph.

  Line 145–146: "We then evaluate the performance of FootNet in predicting individual footprints for the two regions. …"

- Fig. 3F: I was a little confused about Fig. 3F. Are the individual footprints summed before being plotted in this figure? I.e., does this figure compare the log sum of each predicted footprint against the log sum of each true footprint? Alternately, is each individual model grid box from each footprint a different point on this plot? I imagine that the former plot would show less noise and a higher correlation coefficient whereas the latter plot would show more noise and a lower correlation coefficient. I would recommend clarifying how this figure is constructed.

Figure 3F shows the two-dimensional histogram of all of the individual footprint values simulated by STILT and the corresponding predictions made by FootNet in the test data set. Namely speaking, the 2D histogram includes all the 400x400 log-transformed footprint values from each of the 20000x15%=3000 samples from the SF Bay Area and the Barnett Shale region. We have modified the figure caption to make it clear.

---

## Author Response (AR2)

We thank the editor for the comments. We feel that they have helped improve the manuscript. Our response is color-coded in blue.

Can the random training/test split produce samples in each subset that are close in space and time so that the corresponding footprints are very similar? Since such correlations between training and test samples would lead to an overestimation of test performance and hence generalisability, can this be ruled out?

We thank the editor for this comment. In the first version of FootNet, we attempted to address this issue by using footprints from two regions with different meteorology. As such, this version of FootNet is generalizable to both simple (Barnett Shale) and complex (SF Bay Area) regions. You are correct that the performance may be optimistic in this work. The focus of this manuscript is the development of a proof of concept: FootNet v1. We have a companion paper that uses this model within a flux inversion system to test the viability of quantifying emissions using a machine learning surrogate model (Dadheech et al., ACP under review). We also have preliminary results from ongoing work generalizing the FootNet model to any region. The preliminary results indicate comparable performance to what is shown here, but that analysis is very much ongoing.

In terms of peer-review, we submitted this manuscript as the proof of concept that we can effectively emulate the source-receptor relationship under simple and complex cases, a second paper showing that the emulator can be used within a flux inversion framework, and aim to have a third manuscript generalizing this framework.

Line 197: "Due to the computational cost required by the generation of high-resolution footprints, we only included footprints generated from previous studies for the two locations in training version 1.0 of FootNet. We are actively generating new footprints at 1 km from a broader region to further improve the emulator's performance, especially in regions with different meteorological conditions from the two locations used in this study (Dadheech et al., under review). Generalizing this source-receptor emulator to other regions is being tackled in the next version of FootNet."

Code and data availability section: Please add the archives
https://zenodo.org/records/12752655
https://zenodo.org/records/12803617
https://zenodo.org/records/12803736
https://zenodo.org/records/12803855
Thank you. The archives are now added in the revised manuscript.

Line 78: In the units of H the per area seems to be missing. Are the units pbb / (nmol m-2 s-1) per grid cell (i.e. per km2)? In general, I think that the Footprint H, being the central quantity of this study, deserves to be introduced with an equation.

Thank you for the comment. The footprints can be thought of as the mapping between emissions and concentrations. The concentrations are the product of the spatial map of emissions (in units of nmol/m$^2$/s) multiplied by the footprint. When a spatial map of gridded emissions is multiplied by the spatial footprints, the resulting quantity is a concentration in units of ppb. We have added a paragraph introducing footprint and its unit in the paper.

Line 64: "The output of FootNet is a source-receptor relationship (i.e., footprint, **H**), which represents the sensitivity of atmospheric concentrations at a receptor site to emissions upwind of the receptor. This relationship between the measured concentrations and the emissions in the upwind area can be formulated as

$$\mathbf{y} = \mathbf{Hx} + \mathbf{b}$$

where **y** represents the measured concentration, **x** is the emission fluxes in a domain around the measurement location, and b is the background concentration upwind of the domain. The units of **y** and **x** can be expressed as dry air mixing ratio (ppb) and flux rates (nmol m$^{-2}$ s$^{-1}$), respectively (Lin et al., 2003). The source-receptor relationships, $\mathbf{H} = \partial\mathbf{y}/\partial\mathbf{x}$, therefore have units of ppb / (nmol m$^{-2}$ s$^{-1}$)."

Line 89: Is the target also scaled?
The target is log-transformed to reduce the skewness of the distribution of footprint values. This information is added in the paragraph.

Line 78: "The output of FootNet is measurement footprints and is transformed by the natural logarithm function to reduce the skewness of the distribution of footprint values."

Line 113 and Figures: Does "log" mean "ln" or "log_10"? Could you please clarify the units in the log(H) plots, what are the units of H before applying the log?
"Log" means "natural logarithm". This information is corrected throughout the paper. The unit of H is clarified in all the log(H) plots now.

Line 120, Eq. (1): Please explicitly introduce the |.| notation.
The |.| notation is now introduced in line 111.

Line 111: "The absolute value bars ($|\cdot|$) here refer to the area of a region. Specifically, $|Y \cap \hat{Y}|$ represents the area of the region where both the truth and FootNet predictions show non-zero footprints. Similarly, $|Y \cup \hat{Y}|$ calculates the area of the region where either the truth or FootNet predictions show non-zero footprints."

---

## Author Response (AR3)

We thank the editor for the additional comments. Our response is color-coded in blue.

I understand that this study is focused on proof of concept for the two example regions. However, given the random selection of the test dataset, can you be sure that the model really does generalise well to other meteorological conditions in these two regions (not other regions)? For example, the simultaneous footprints in the files

X_Y_-97.94Ex33.03N_2013102000.npz

X_Y_-97.98Ex33.05N_2013102000.npz

are for receptors that are close together, so the corresponding footprints and inputs are very similar. My understanding is that one of them may have been in the training dataset and the other in the test dataset (if this doesn't apply to these two footprints, there are certainly other examples). In this case, the model only needs to reproduce the footprint in the training dataset during testing, which would lead to good test results even for a non-generalising model. Could you discuss whether or not this might be a problem here? It could be tested and avoided by, for example, splitting the dataset into different time periods rather than randomly.

We appreciate the editor for this comment. Yes, we acknowledge that similarities between samples are hard to fully avoid during the construction of version 1 of FootNet, even if the construction of the training and test data sets is done by splitting data by time periods rather than random selection. We tested the training split proposed by the editor using the training data set, and indeed found degradation in the performance.

We also evaluate the training split strategy proposed by the editor in our more recent updates of the model. We find little-to-no difference in performance, indicating that when the data volume is largely increased the generalizability of FootNet could be improved.

We have added the following line to the text to emphasize the results demonstrated from the construction of FootNet version 1 could be affected by different split strategies of the training data set:

Line 138: "However, it is worth mentioning here that we find some performance degradation using an alternative splitting of the data based on different time periods. Because the training data set used to construct version 1 of FootNet has a relatively small size, similarities between samples are hard to fully avoid by randomly selecting training data samples, which could lead to generalizability issues when using FootNet version 1 over regions and time periods too different from the training data set. This generalizability issue could be largely mitigated by increasing the volume of the training data set in the future (Dadheech et al., 2024)."

One minor comment: in footnet.py, the footprint transformed by the protocol seems to be shifted by 20, which is not mentioned in the manuscript and corresponds to a scaling of the footprint by exp(20).

Thank you. The offset is applied to filter and remove too small footprint values generated by the physics-based model. We have now mentioned this in the manuscript.

Line 80: "The transformed footprints are filtered to remove values smaller than -20 and then shifted by +20, corresponding to a scaling of the raw footprints by exp(20)."